# Rare Etiologies of Upper Gastrointestinal Bleeding: A Narrative Review

**DOI:** 10.3390/jcm14144972

**Published:** 2025-07-14

**Authors:** Ion Dina, Maria Nedelcu, Claudia Georgeta Iacobescu, Ion Daniel Baboi, Alice Lavinia Bălăceanu

**Affiliations:** 1Semiology Department, “Carol Davila” University of Medicine and Pharmacy, 020021 Bucharest, Romania; ion.dina@umfcd.ro (I.D.); alice.balaceanu@umfcd.ro (A.L.B.); 2Gastroenterology Department, Clinical and Emergency Hospital “Sf. Ioan”, 042122 Bucharest, Romania; iacobescu_clodi@yahoo.com; 3Internal Medicine Department, “Carol Davila” University of Medicine and Pharmacy, 020021 Bucharest, Romania; 4Internal Medicine Department, Clinical and Emergency Hospital “Sf. Ioan”, 042122 Bucharest, Romania

**Keywords:** upper gastrointestinal bleeding, rare cause, diagnostic process, narrative review, research methodology, gastric metastases, pyloric gland adenoma, gastrointestinal stromal tumor

## Abstract

Rare presentations are surprising and may disturb the day-to-day routine of a medical unit; however, they are expected (not as individual entities, but as a group of “uncommon causes”). While reviewing the literature in relation to three clinical cases of upper gastrointestinal bleeding (UGIB) encountered in our institution—gastric metastases of breast cancer (GMB), pyloric gland adenoma, and gastrointestinal stromal tumor (GIST)—we identified seven and 29 case reports for the first two entities, and over 100 publications addressing GIST. This prompted a shift in focus from novel reporting to diagnostic contextualization. We found it difficult to obtain an overview of the spectrum of UGIB etiologies, as most publications refer to a few individual entities or to a subgroup of rare causes. The narrative review we conducted arose from this particular research methodology. Based on a broad literature search, UGIB etiologies were organized in five categories (lesions of the mucosa, neoplasms, vascular causes, bleeding predisposition, and external sources of bleeding). In the management of patients with UGIB, the underlying etiology deviates from the classic peptic ulcer disease/esophageal varices dyad in approximately half of the cases. This underscores the need for heightened clinical vigilance, particularly in complex scenarios, where endoscopic findings, imaging results, and histopathological interpretations may be unexpected or prone to misinterpretation. As an illustration, we conducted two systematic reviews of case reports of bleeding GMB and PGA. Our findings support a proactive diagnostic and research mindset and advocate for improved awareness of uncommon UGIB etiologies.

## 1. Introduction

Between 1980 and 2020, more than 12,500 publications referencing adult gastrointestinal bleeding in their titles were indexed in the Web of Science (WOS) database (by Clarivate Analytics), and it is estimated that the volume of medical knowledge doubles approximately every five years [1]. Among these studies, case reports and reviews focusing on atypical presentations of upper gastrointestinal bleeding (UGIB) may often fall outside the scope of standard guidelines. Yet, they may hold particular relevance for everyday clinical decision-making.

Unusual presentations are regularly encountered in all medical settings. Peptic ulcer disease (PUD) and esophageal varices (EV) are the most commonly suspected causes of UGIB; however, together, they account for less than half of the cases (32–36% and 11%, respectively) [2]. Even those common causes can occasionally result from rare underlying conditions [3,4]. Notably, patients with idiopathic ulcers experience higher rates of rebleeding and mortality compared with those with clearly defined etiologies [5].

This narrative review emerged from the need to contextualize and integrate the rare cases of UGIB encountered in our medical unit within a broader framework. In modern evidence-based medicine, the physician’s expertise lies in integrating the multitude of available clinical guidelines and research findings with patient preferences and the specific resource constraints of the hospital or individual case [6]. A comprehensive understanding of rare etiologies of upper gastrointestinal bleeding and the potential diagnostic pitfalls can enhance interdisciplinary communication and support a more targeted application of diagnostic tools. Narrative reviews are well suited to addressing this topic more broadly and allow the researcher to use an exploratory strategy [7]. To support this contextualization, we conducted a focused systematic search of published case reports for three of the rare UGIB etiologies encountered in our unit (gastric metastases of breast cancer, pyloric gland adenomas (PGAs), and gastrointestinal stromal tumors (GISTs)).

## 2. Materials and Methods

Anchored in three clinical cases with rare proliferative UGIB etiologies encountered in our unit, we conducted a systematic review of case reports in the WOS for cases of gastric metastases of breast cancer, PGA, and GIST causing UGIB. Only one database was systematically searched, which limits the comprehensiveness of the findings and may exclude relevant case reports published in other sources. This review should be interpreted as a focused, systematic exploratory synthesis rather than an exhaustive systematic review. The search was performed in May 2025, including case reports of overt UGIB of the mentioned causes published between January 2014 and May 2025. Cases of patients presenting only with unexplained anemia or positive occult blood tests were excluded. The search terms were in the English language, but articles written in other languages were not excluded, as the identified languages were familiar to the authors. The search terms were (((TI = (bleed)) OR TI = (hemorrhage)) OR TI = (ugib)) OR TI = (anemia) for the topic of UGIB, fol in addition to the following:-((TI = (breast)) OR TI = (Mammary)) and ((((((TI = (gastrointestinal)) OR TI = (duodenum)) OR TI = (gastric)) OR TI = (digestive)) OR TI = (esophagus)) OR TI = (stomach)) for GI metastases of breast cancer;-(((TI = (brunner)) OR TI = (brunneroma)) OR TI = (hamartoma)) OR TI = (pyloric gland adenoma) for pyloric gland adenomas;-((TI = (gastrointestinal stromal)) OR TI = (gist)) for gastrointestinal stromal tumors.

After one of the authors screened the titles and the abstracts of the results, studies that were not case reports or did not respect the intended topic (i.e., covered other clinical entities) were excluded. The volume of the existing literature identified for GISTs manifesting as UGIB prompted us to forgo a case-based systematic review in favor of a narrative synthesis of relevant insights. For the case reports of gastric metastases of breast cancer, we collected data on tumor histopathological identification, macroscopic lesion characteristics, and patient-related variables, including age, sex, hemoglobin level, prior knowledge of the neoplastic disease (i.e., presence of a former cancer diagnosis), management, and outcome. For the case reports of pyloric gland adenoma (PGA), we collected data on tumor location and size, as well as patient characteristics, including sex, age, hemoglobin level, and treatment approach.

As the review progressed, it became evident that proliferative causes intersect with other etiologies of UGIB, and the aim of the research extended to offering a broader view of the etiological spectrum encountered in clinical practice. We conducted a focused narrative review of UGIB etiologies using PubMed, WOS, and Google Scholar, using an exploratory, iterative search strategy guided by clinical relevance, as per SANRA recommendations [7]. Original articles, reviews, and case reports that focused on etiologies of UGIB, including reports of gastrointestinal manifestations of systemic conditions, were included until thematic saturation was reached, and it was apparent that no significantly new etiologies were addressed. Although not exhaustive, this method allowed the collection of a broader picture of the underlying conditions.

## 3. Results and Discussion

### 3.1. Contextualising the Clinical Cases; Explaining the Aim and the Limitations in the Context of the Results

The experience of our medical unit included several incidents of rare underlying causes of UGIB. In the process of reviewing the literature for three individual case reports (gastric metastases of breast cancer, PGA and GIST), we identified 7 case reports published in the previous 10 years on bleeding gastric metastases of breast cancer and 29 on bleeding PGA; on bleeding GISTs, we identified 105 general articles, including a 2025 systematic review and meta-analysis of the prognosis [8], and a 2018 comprehensive review of the management of hemorrhage [9]. Therefore, as mentioned above, a new case report and synthesis were not justified for GISTs. The main overall limitation of the included case reports was the absence of long-term follow-up information in several studies.

Although rare cases are surprising and may disturb the day-to-day routine of a medical unit, they are expected (not as individual entities, but as a group of “uncommon causes”). We found it difficult to obtain an overview of possible rare conditions that could cause UGIB in our clinical practice, as most publications referred to a few individual entities or to a subgroup of rare causes. The narrative review we conducted arose from the research methodology that we applied for the contextualization of our case reports; since one of the hallmarks of case reporting is to recognize the novelty of the case, the methodology of contextualization is as valuable as the reporting of the unexpected findings [10]. This article is a demonstrative exercise of this methodology.

As a consequence, this review is not intended as an exhaustive or systematic synthesis of all rare etiologies of upper gastrointestinal bleeding (UGIB) but rather as an exploration that highlights diagnostic and management challenges. The analysis of case reports included in the review reflects a heterogeneous body of literature, with considerable variability in the level of clinical detail, diagnostic workup, and outcome documentation. Moreover, the nature of case report publishing inherently favors unusual or dramatic presentations, which may overrepresent the frequency and perceived complexity of certain conditions, but “the unexpected is where discovery begins” [10]. As such, the present work did not aim to inform guideline revisions or establish generalizable clinical pathways. Instead, its purpose was to underscore potential diagnostic pitfalls and encourage a proactive, critical approach to clinical decision-making, particularly in atypical or unclear UGIB presentations. The point to be made is that diagnostic errors in encounters with rare presentations may highlight generalizable weak points (as the rate of misdiagnosis of common conditions is not to be ignored [11]). While this review does not adhere to systematic review protocols, it offers a structured overview of unexpected findings that can arise during the patient management process, reinforcing the importance of clinical vigilance in daily practice.

### 3.2. Overview of UGIB Etiologies—A Narrative Review

The literature offers extensive information on rare UGIB etiologies, including several reviews [12,13,14]. Figure 1 [2,3,4,12,13,14,15,16,17,18,19,20,21,22,23,24,25,26,27,28,29,30,31,32,33,34,35,36,37,38,39,40] presents an overview that integrates these reviews with other sources, such as reports on rheumatic diseases associated with UGIB [17], as well as a range of etiologies we identified through case reports systematization. 

A useful resource that offers a systematic and detailed approach to most of the causes of UGIB, highlighting management particularities for many uncommon conditions, is the of *Practical Approach to Diagnosis and Management of GIB* [41]. We grouped the etiologies infive classes (lesions of the mucosa, neoplasms, vascular causes, external bleeding sources, and bleeding predisposition, which could severely amplify preexisting hemorrhagic risk). Each of the outlined categories encompasses common and rare entities (although external bleeding causes tend to be rare findings). For example, while ulcers and portal hypertension are the most common sources of UGIB, there are just a few cases of ulcerations caused by *Sarcina ventriculi* infection reported in the literature [13], and there are multiple other rare subjacent causes, such as Zollinger–Ellison syndrome [42,43] or pseudocirrhosis [44,45,46].

However, the etiology may fluctuate across diagnostic categories, and an apparent definitive classification may sometimes obscure another cause. For example, a neoplasm could be associated with a systemic disorder (such as neurofibromatosis type 1 in the case of GIST [31]) or cause a coagulopathy; peptic ulcer disease may result from a gastrinoma [3]; esophageal varices may be secondary metastatic cancer or mediastinal fibrosis [4,24]; visceral artery aneurysms can arise from chronic pancreatitis [14]; and Mallory–Weiss syndrome could reveal a methanol intoxication [47]. As a separate mention, several vitamin deficiencies have been reported to aggravate or cause UGIB [30,48].

The UGIB patient management consists of stabilization and pre-endoscopic measures, identification of the source of bleeding, hemostasis, and the definitive management that follows the final diagnosis [41].

The pre-endoscopic measures are PUD and EV oriented [2]. However, even pre-endoscopically, a high clinical suspicion and a careful patient history can influence the management strategy (for example, the identification of a bleeding disorder caused by medication or comorbidities, searching for the cause of emesis in a suspected Mallory–Weiss case, opting for a CT scan instead of endoscopy in the case of suspected aorto-enteric fistula; laboratory evaluations could point to a pancreatic or biliary source of the bleeding that even endoscopy proves nondiagnostic) [41].

Endoscopy identifies approximately 80–90% of UGIB lesions [41]. Elusive lesions could be identified through angiography or angiographic imaging. However, aorto-enteric fistula, for example, has traditionally had low detection rates and, sometimes, only a CT scan could provide preoperative diagnosis [41]. In other cases, however, the CT scan could provide significant false-negative results, which must be taken into consideration, especially when investigating anemia [49,50,51]. In some cases, endoscopies have a limited role: they could exclude other sources of bleeding, but they can be misinterpreted, for example, in the case of ruptured visceral artery aneurysms (VAAs), gastric submucosal arterial collaterals (GSACs), hemosuccus pancreaticus, or hemobilia. Standard hemostatic techniques have little role in the treatment of VAA, and misidentifying GSAC as gastric varices and proceeding with endoscopic glue injection before angiographic imaging can result in severe complications [14].

In a 2004 Scottish study of 30 patients, 56% of esophageal cancers were misinterpreted as esophagitis or benign stricture in recent endoscopies, and 71.4% of gastric cancers as gastritis, ulcers, or inconclusive lesions. In 22 cases, diagnostic opportunities were clearly missed—one third due to pathologist error, the rest to endoscopic misinterpretation [52]. While diagnostic accuracy has since improved, recognizing that every step of the diagnostic process is fallible—and maintaining epistemic humility—is essential for continued progress.

It is important to acknowledge that histopathological diagnosis is often established only after the patient has been discharged. Furthermore, more than a quarter of probe identification errors are identified by clinicians, not by other histopathologists [53]. Therefore, the clinician’s ability to apply clinical expertise in forming a pre-biopsy differential diagnosis is essential—it can help avoid missed biopsy opportunities and misinterpretation of false-negative results and guide appropriate next steps in cases with inconclusive findings.

### 3.3. Rare (Proliferative) Etiologies: Three Case-Anchored Systematic Reviews with Diagnostic Management Considerations

To illustrate the practical significance of infrequent etiologies of upper gastrointestinal bleeding, we present three clinical cases encountered in our institution, each associated with a proliferative lesion: gastric metastasis of breast cancer, PGA, and GIST. We conducted a focused, systematic review of case reports indexed in the WOS. These mini-reviews contextualize the rarity, presentation, diagnostic approach, and outcomes associated with these pathologies.

Upper gastrointestinal tumors are estimated to cause 1–5% of all acute UGIB [40]. This is of clinical significance since it has been estimated that one in 26 people from high-human-development countries would be diagnosed with stomach or esophageal cancers during their lifetime (metastatic cancer only adding to those numbers). Studies indicate that more than half of those malignant tumors could have UGIB as the presenting symptom [40,41].

#### 3.3.1. Gastric Metastases from Breast Cancer

The illustrated patient (Box 1) was diagnosed with metastatic breast cancer after an emergency consult for active hematemesis. In just three months, she presented with rebleeding and gastric outlet obstruction, which required an expandable stent. Although breast cancer is one of the most common primary sites for gastric metastases [31], it is a rare cause of UGIB or gastric outlet obstruction, especially as the presenting symptom in a previously undiagnosed patient.

Box 1Case report of a gastric metastasis of breast cancer presenting with upper gastrointestinal bleeding.A 72-year-old female without any known medical history
presented with active hematemesis and moderate anemia. Emergency upper
endoscopy revealed a Forrest IIa ulcer with a visible vessel; multiple
hemoclips were applied. Biopsies taken during reevaluation showed malignant
infiltration of the lamina propria by “signet ring”-like cells, with lobular breast cancer immunohistochemistry (Figure 2). The thoracoabdominal CT scan showed a 40 mm breast formation with nipple retraction, left subclavian vein thrombosis,
osteocondensations suggestive of bone metastases, and possible peritoneal carcinomatosis. The patient was discharged hemodynamically stable.Three months later, she presented again with hematemesis,
in a poor general condition. Endoscopy showed retained food debris and a
circumferential stenosing antral mass obstructing duodenal access, with
fibrin-covered superficial erosions and no active bleeding. After cardiologic
and oncologic examination, a new gastroscopy was performed under sedation and
mechanical ventilation, and an expandable stent was placed from the antral
region to the first part of the duodenum. The patient was discharged afterward,
hemodynamically stable.

**Figure 2 jcm-14-04972-f002:**
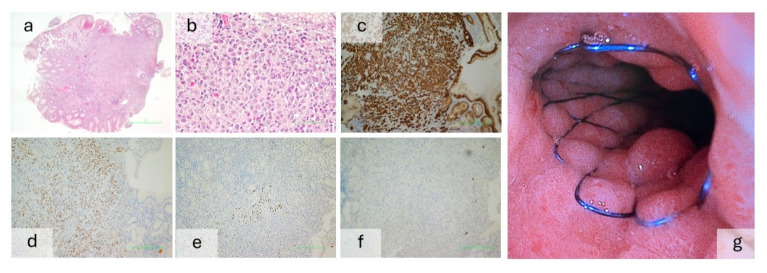
(**a**–**f**): Histopatology and immunohistochemistry of the gastric metastasis from the breast. (**a**) Hematoxilin-eosin, 4×, diffuse proliferation of discohesive cells in lamina propria. (**b**) Hematoxilin-eosin, 40×, diffuse proliferation of discohesive cells in lamina propria. (**c**) Immunohistochemistry–MCK, 10×, positive cytoplasmatic perimembranous. (**d**) Immunohistochemistry–ER, 85% nucleary positive (**e**) Immunohistochemistry–PR, 10×, 10% nucleary positive. (**f**) Immunohistochemistry–C-erb B2/ Her2 Neu, 10×, negative. (**g**) Expandable stent placed in the antrum and the first part of the duodenum. The patient was also GATA3-positive perinucleary—not shown.

In a 2017 literature review of 77 isolated cases in the literature, breast cancer was diagnosed after a consultation for gastrointestinal symptoms in 17.9% of cases, and 23.1% of patients newly diagnosed for breast cancer had gastric metastases at the moment of diagnosis. A total of 7.7% of all patients had bleeding as the main symptom, and 62.8% of all patients had other metastases at the moment of stomach involvement [54]. However, it is worth mentioning that the presence of gastric metastasis in breast cancer patients tends to be significantly higher in autopsy reports than in clinical retrospective studies [55], which may occur due to the non-specific presentations (the most common symptom being abdominal pain), lack of radiological precision in diagnosis, or misinterpretation of endoscopic features. For example, in a 2019 case report, a breast cancer metastasis causing gastric outlet obstruction was initially discharged with the diagnosis of chronic gastritis. In a recent case series, one patient was treated for primary gastric cancer for 1.5 years before the correct diagnosis [55].

The systematic review of case reports (described in the Methodology section) resulted in 11 publications, of which seven were on the selected topic and were summarized in Table 1. All case reports were on female patients. None of these cases presented with both hemorrhage and obstruction, although it has been mentioned in older reports [56]. All patients had other metastases, and only two were oblivious of the cancer diagnosis [57,58]. Notably, one of the patients had regular mammographies and negative CT and MRI scans of the breasts: inframammary skin induration and UGIB were the only signs of disease, although the cancer had already disseminated in multiple regions.

The cause of the bleeding was a metastatic lesion in most patients, but one patient needed 13 hemostatic sessions for gastric antral vascular ectasia (GAVE), likely induced by Everolimus [59] and another one had esophageal varices as a consequence of liver metastases [60], invasive ductal carcinoma being one of the main etiologies of pseudocirrhosis [61]. For the record, another possible cause of UGIB for breast cancer patients is “downhill” varices. Considering that coagulopathies are more frequent in patients with malignancies, UGIBs of breast cancer patients could be caused by all five categories mentioned in the above narrative review [62].

In two cases, the anemia of the patient was documented 6 months prior to the GI metastasis diagnosis. As previously reported, these patients had a high risk of rebleeding and an unfavorable prognosis (the median survival rate in the review of Liang Xu et al. was 10.5 months).

The presented cases plead for a few management considerations. There are multiple mechanisms that could cause UGIB in breast cancer patients. GI metastases are to be considered for breast cancer patients with anemia, keeping in mind the possible unreliability of the CT in the identification of such lesions [50,57,63], especially considering the fact that chemostatic agents could aggravate the risk of bleeding [64]. An understanding of the spectrum of possibilities and a close collaboration in the multidisciplinary team could reduce the number of misdiagnosed patients. Furthermore, given the terminal disease these patients tend to have, the availability of palliative care services is of great importance.

**Table 1 jcm-14-04972-t001:** Case reports of breast-cancer patients presenting with UGIB identified in the Web of Science Core Collection in the 2014–2025 interval. NA = not available. Hb = hemoglobin level. GAVE = gastric antral vascular ectasia.

Author (Year)	Age	Former Cancer Diagnosis	Tumor Identification	Hb (g/dL)	Lesion	Management	Outcome
Delungahawatta (2023) [50]	74	Yes	CK7, GATA3 positive	6.8	Non-bleeding, friable nodular gastric mucosa	Palliative care	NA
Delsa (2022) [60]	64	Yes	HER-2neu and HR positive	7.4	Grade 3 esophageal varices with active bleeding	Endoscopic variceal band ligation.	Rebleeding and loss of consciousness in the following week, followed by the discovery of brain metastases and exitus.
Liu (2020) [64]	82	Yes	Phyllodes tumor	8.8–5.4	Protruding, gastric tumor with punctate bleeding	Gastrostomy with excision of gastric tumor	No rebleeding; refused chemotherapy; brain metastases were found, and she died 2 months later.
Ulanja (2018) [57]	75	No	ER/PR, GATA3, mammoglobin positive, HER2 and E-cadherin negative	7.6	Erosive gastritis and duodenitis and non-obstructing Schatzki ring at the gastroesophageal junction	Conservative management of the bleeding.	NA
Khan (2017) [58]	56	No	NA; tumor onfimed by DNA microarray	5.2	Multiple ulcerated gastric mucosal nodules	NA	The patient died after the start of chemotherapy.
Choi (2017) [63]	44	Yes	Phyllodes tumor	6.7	Gastric mass (approximately 7 cm in diameter) with active bleeding and several related masses	Endoscopic hemostasis with cauterization.	Rebleeding in 2 days and 1 month.
Assi (2014) [59]	48	Yes	ER/PR positive, HER2 negative	4.5	GAVE	Endoscopic hemostasis with cyanoacrylate spray and argon plasma coagulation.	Multiple rebleeding episodes that ended after the cessation of Everolimus.

#### 3.3.2. Pyloric Gland Adenoma

Small bowel tumors account for a few UGIB cases but require multiple interventions until the diagnosis is made, and come with a great burden to the healthcare system [41].

Pyloric gland adenomas (PGAs) of the small intestine, commonly named Brunner’s gland adenomas, are polyps consisting of dysplastic glandular epithelium with pyloric gland differentiation, with a small risk of progression to adenocarcinoma [31]. They are found in 0.01–0.07% of the patients examined endoscopically, and although they are mainly asymptomatic or cause non-specific abdominal discomfort, UGIB is a major reason for patient presentation [65].

Like many other quite rare pathologies, they could be misdiagnosed. In a 2018 case report, a 48-year-old male underwent pancreaticoduodenectomy for a 4 cm lesion of the pancreas head that turned out to be a PGA [66]. In a 2025 case report, a 56-year-old male presented with an unintentional 10 kg weight loss due to a duodenal stricture that was histopathologically recognized as a Brunner adenoma only after gastrojejunostomy [67].

The systematic review of case reports (described in the Methodology section) resulted in 46 publications, and the relevant 29 articles were summarized in Table 2. The review highlighted that delays caused by multiple negative small-sample biopsies before resection are common [68]. Furthermore, some of those patients presented multiple times with unexplained abdominal discomfort before the final diagnosis or had uncommon presentations, such as intussusception [69,70]. Although delays are more forgiving in the case of benign lesions, they are significant in the physician’s mental and physical resources used for the differential diagnosis. In some of the patients, the delays translated into the symptoms of moderately severe anemia (the average hemoglobin levels in the reviewed cases, presented in Table 1, were 7.76 g/dL (SD = 2.50), with a minimum of 4.19 g/dL [71]).

Another particularity in some of the reviewed cases was the reported absence of acid secretion or *Helicobacter pylori* infection as a causal factor, which was also the case for our patient (Box 2) [72,73,74]. Furthermore, one of the cases that presented with UGIB was a lesion smaller than 5 mm, although hemorrhage is expected in large polyps [75].

Box 2Case report of a pyloric gland adenoma presenting with upper gastrointestinal bleeding.A previously healthy 48-year-old male presented to the
emergency department for coffee-ground emesis and melena. The patient had
moderate to severe anemia, and the test for Helicobacter pylori antibody test
was negative. Endoscopy showed reflux esophagitis and a 5 cm pedunculated
polypoid formation in the bulbar region that was extending through the
pylorus into the gastric antrum. The lesion was ulcerated at both its stalk
and tip and was also identifiable on the CT scan. A repeat endoscopy was
performed for polypectomy using endoloop ligation, without the excision of
the polyp’s stalk (Figure 3). Histopathologically, the lesion was reported to be
lined by intestinal-type epithelium, showing a lobular submucosal
proliferation of Brunner glands, adipose tissue, fibrous connective septa and
smooth muscle tissue, with minimal chronic inflammatory infiltrate,
interpreted as a Brunner gland adenoma. The patient was discharged in a
stable condition.

**Figure 3 jcm-14-04972-f003:**
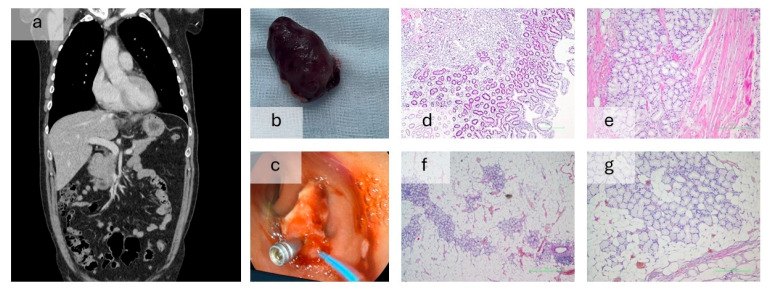
Images obtained during the investigation of the Brunner’s gland adenoma. (**a**) Contrast-enhanced CT scan. (**b**) The resected piece, sent for biopsy. (**c**) Endoscopic removal of the formation. (**d**–**g**) Hematoxilin-eosin coloration of the resected piece. (**d**) 4×, duodenal mucosa with submucosal proliferation of Brunner glands. (**e**) 10×, Brunner gland lobules with muscular tissue. (**f**) 4×, Brunner gland lobules with adipose tissue. (**g**) 10×, Brunner gland lobules with adipose tissue and fibroconnective septa.

Although endoscopic features combined with biopsy would be the most intuitive approach, endoscopic ultrasonography (EUS) is considered the most accurate for assessing the origin of the polyp, and contrast CT scan could help distinguish it from other entities. Endoscopic resection is usually preferred to surgery for stalked lesions smaller than 2 cm or asymptomatic. However, there were cases of polyps up to 12 cm resected endoscopically [74,76,77]. The endoscopic approach to large polyps was sometimes complicated by the need for repeated procedures [78,79,80], incomplete resections [81] or eventual surgical intervention, either due to failed endoscopic attempts [82] or patient preference after three previous endoscopic procedures [80]. There was a report of a successful combined simultaneous endoscopic and laparoscopic approach [83].

**Table 2 jcm-14-04972-t002:** Case reports of patients with Brunner gland adenoma identified in the Web of Science Core Collection in the 2014–2025 interval. NA = not available. Hb = hemoglobin level. DI, II, III, IV: parts of the duodenum.

First Author (Year)	Sex	Age	Hb (g/dL)	Location	Size (mm)	Treatment
Russel (2024) [84]	Female	47	5.6	DI-DII	50	Endoscopic resection
Gonzalez (2024) [78]	Male	25	6.4	pylorus-DI	60–80	Endoscopic resection in three sessions
Jourdain (2024) [85]	Male	88	8.2	NA	NA	Endoscopic resection
Jaber (2024) [86]	Female	70	5.4	DII	21	Endoscopic resection
Rubin (2024) [87]	Female	88	6.5	DI-pilorus	50	Endoscopic resection
Mendes (2024) [88]	Male	78	NA	DI	40	Endoscopic resection
Yan (2024) [72]	Female	31	9.6	DI	40	Endoscopic resection
Al Hariri (2023) [75]	Male	25	6.3	DI	<5	Conservative
Cruz (2023) [89]	Male	66	12.3	DI	40	Endoscopic resection
Cai (2023) [90]	Male	56	12.4	DI-DII	NA	Endoscopic resection
Marano (2023) [83]	Female	41	NA	DI	50	Combined laparoscopic-endoscopic approach
Shaheen (2023) [91]	Male	52	5.9	DI-DII	39	Laparoscopic resection
Okutomi (2021) [92]	Female	33	4.9	DI	70	Surgical resection
Nguyen (2021) [69]	Male	26	7.2	NA	NA	Conservative
Dhali (2021) [82]	Female	44	7	DI	25	Endoscopic resection failed, and surgical resection was necessary
Ma (2020) [93]	Female	81	6.6	DII	60–70	Endoscopic resection
Abushamma (2020) [94]	Female	58	7.9	Pylorus	40	Endoscopic resection
Bakheet (2020) [73]	Male	55	9.4	DI	32	Endoscopic resection
Kamani (2020) [95]	Male	40	6	DII-DIV	60	Surgical resection
Rana (2019) [74]	Male	76	11.9	DII	100–120	Endoscopic resection
Herreras (2019) [96]	Female	57	14.1	DI	8	Endoscopic resection
Bancila (2018) [97]	Male	76	NA	DI	NA	Endoscopic resection
Ulusahin (2018) [98]	Female	56	6	DII	40	Surgical resection
Frenkel (2017) [71]	Male	54	4.19	DI	80	Surgical resection
Chilukuri (2017) [99]	Female	81	6.1	DI-DII	NA	Endoscopic resection
Sorleto (2017) [80]	Female	47	8.2	DI	60	Three endoscopic resection sessions followed by surgical resection of the residue
Lee (2016) [79]	Male	30	9.7	DI-DII	93	Endoscopic resection in two sessions
Hsu (2016) [100]	Male	66	7.8	Antrum	NA	Endoscopic resection
Bernard (2015) [81]	Female	81	6.4	NA	55	Incomplete endoscopic resection
Liu (2014) [70]	Female	49	7.6	DII	60	Surgical resection

#### 3.3.3. Gastrointestinal Stromal Tumor

GISTs are considered rare neoplasms (reported 1.1–1.5 cases per 100.000 person-years in a Scandinavian study [31]), but ones smaller than 1 cm lesions are probably more frequent. The search intended for a systematic review of case reports, described in the Methodology section, identified 105 articles, including a 2025 systematic review and meta-analysis of the prognosis of GIST patients with UGIB [8] and a 2018 comprehensive review of the management of hemorrhage in GIST [9]; therefore, we shifted our focus from a systematic review to a diagnostic contextualization.

GISTs have a relatively high risk of malignancy (a quarter of gastric GISTs are clinically malignant), and UGIB is one of the common ways of presentation [31]. Bleeding GISTs are associated with an unfavorable prognosis and are at a greater risk of recurrence [9]. They can appear in varied forms, with sizes from a few millimeters to up to 20 cm, as ulcerations, intraluminal masses, or lesions with cystic or necrotic changes. The unspecific presentation translates to a lengthy diagnosis process: in a real-world data study in the Netherlands published in 2025, GIST patients had an average of six general practitioner consults in the 12 months prior to diagnosis compared with three in the control population [101].

Our patient, presented in Box 3, had a history of alcohol abuse, and his presentation with a gastric ulcer could have been interpreted as a classic ulcer. However, the availability of EUS and the decision to evaluate the lesion histopathologically led to the correct diagnosis. The correct management was even more important considering the transfer of the patient to another unit for the treatment of the newly diagnosed tuberculosis, which would have probably delayed the re-evaluation of the lesion.

Box 3Case report of a gastrointestinal stromal tumor presenting with upper gastrointestinal bleeding.A 48-year-old male patient with a history of alcohol
consumption and smoking presented to the Emergency Department with acute
hematemesis and melena. Initial investigations revealed severe anemia. Upper
gastrointestinal endoscopy identified a subcardial gastric ulceration covered
with fibrin, and EUS with elastography identified a low-elasticity, hard
tissue (Figure 4). Histopathological analysis subsequently confirmed the
diagnosis of a gastrointestinal stromal tumor (GIST). Thoracic computed tomography demonstrated multiple nodular pulmonary lesions, and a microscopic
examination of the sputum revealed active tuberculosis. The patient was
transferred to a pneumophthisiology unit with a hemoglobin level of 8.9 g/dL
without signs of active bleeding, with the recommendation of a follow-up
consult for the definitive management of the lesion after discharge.

**Figure 4 jcm-14-04972-f004:**
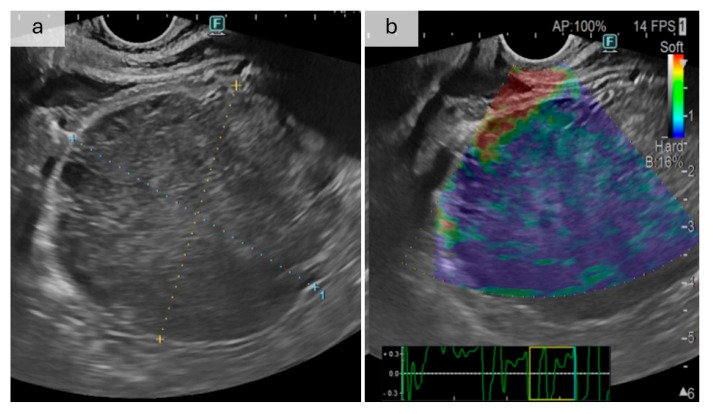
(**a**) Ecoendoscopic images of the gastrointestinal stromal tumor, (**b**) with elastography revealing blue, low-elasticity tissue. The rectangle at the bottom of the figure is a real-time display of the stress applied by the transducer, showing the compression-release and frequency of the push pulses; the yellow box depicts the displayed section. GISTs have heterogeneous presentations that underscore the importance of a structured diagnosis. There are case reports of GIST tumors initially discharged as peptic ulcers [102] or a supposed GIST that turned out to be a peptic ulcer after resection [103]. They could also be misdiagnosed as leiomyomas or leiomyosarcoma before histopathological examination [104], and KIT-negative GISTs have a large spectrum of differential diagnoses even after immunohistochemistry [105]. In a 2023 case report, a 64-year-old woman presented with melena, with undiagnostic endoscopy and MRI findings and a pelvic mass that seemed related to the uterus, and was referred to gynecology. Only after the second presentation with melena, the contrast-enhanced CT scan identified a jejunal GIST [106]. In a Chinese case report, a 65-year-old patient with gastric epithelioid GIST was misdiagnosed and treated for adenocarcinoma for four years [107]. Sometimes, GIST tumors are part of Carney–Stratakis syndrome or associated with neurofibromatosis type 1 [31], which underscores the need for a holistic approach to patients with UGIB.

## 4. Conclusions

The analysis of rare cases implies a contextualization of the novelty of the case by a systematization of the previous literature. “A certain type of case report will (or should) never come back: the droning recital of one case after the other as a lame excuse for an (unstructured) review of the literature [10]”. Since the different types of UGIB etiologies are interconnected, an overview of all (including and starting from rare) cases offers the foundation for this contextualization. Starting from the research of a case series of rare UGIB patients, we needed to proceed to a systematic review of similar case reports and, ultimately, to a narrative review of UGIB etiologies.

In the management of patients with upper gastrointestinal bleeding (UGIB), the underlying etiology deviates from the classic peptic ulcer disease/esophageal varices dyad in approximately half of the cases [2]. This underscores the need for heightened clinical vigilance, particularly in complex scenarios—such as hemodynamically unstable patients, those with preexisting conditions, or presentations occurring in atypical clinical contexts—where endoscopic findings, imaging results, and histopathological interpretations may be unexpected or prone to misinterpretation.

## Figures and Tables

**Figure 1 jcm-14-04972-f001:**
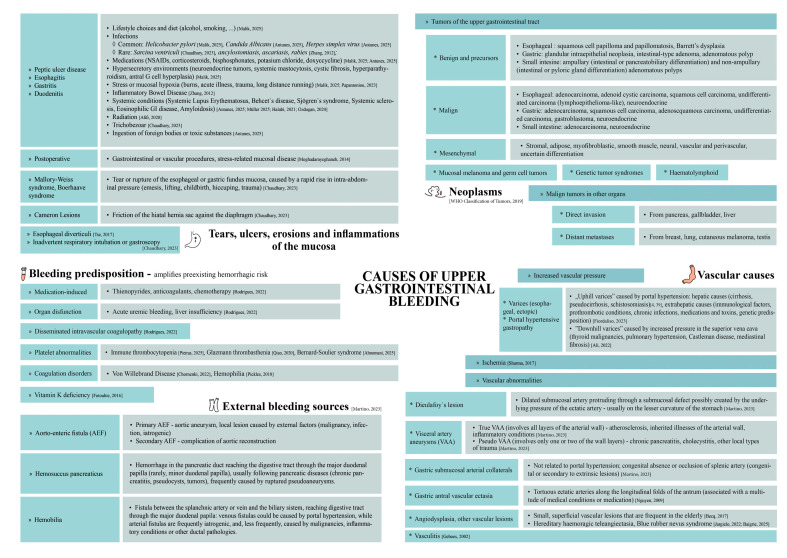
Causes of upper gastrointestinal bleeding—a review including rare causes. Explanations and underlying conditions are listed to the right of the main etiological categories—though not exhaustive, this aims to encourage further exploration of underlying causes through patient history, diagnostic investigations, and clinical features [2,3,4,12,13,14,15,16,17,18,19,20,21,22,23,24,25,26,27,28,29,30,31,32,33,34,35,36,37,38,39,40].

## Data Availability

Data is contained within the article.

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
