# Peer review of "Rare Etiologies of Upper Gastrointestinal Bleeding: A Narrative Review"

_jcm, 2025, doi:10.3390/jcm14144972_

Round 1

Reviewer 1 Report

Comments and Suggestions for Authors

Dear Authors,

Your paper is interesting and my personal suggestions are to make Your narrative review more useful for readers.

"Rare Etiologies of Upper Gastrointestinal Bleeding: A Narrative Review"

  1. General Overview

The paper is a narrative review published in the Journal of Clinical Medicine, focusing on rare causes of upper gastrointestinal bleeding (UGIB). It combines a systematic review of case reports with a broader narrative synthesis to contextualize diagnostic challenges and management strategies for uncommon UGIB etiologies. The study is anchored in three clinical cases encountered by the authors: gastric metastases of breast cancer (GMB), pyloric gland adenoma (PGA), and gastrointestinal stromal tumor (GIST).

  1. Key Sections and Findings
  • Introduction:
    • UGIB is commonly attributed to peptic ulcer disease (PUD) and esophageal varices (EV), but these account for less than half of cases.
    • Rare etiologies, though individually uncommon, collectively represent a significant proportion of UGIB cases, necessitating clinical vigilance.
    • The review emphasizes the importance of integrating rare causes into diagnostic frameworks to avoid misinterpretations and improve patient outcomes.
  • Methods:
    • The authors conducted a systematic review of case reports (2014–2025) for GMB, PGA, and GIST presenting with UGIB, sourced from Web of Science (WOS).
    • A narrative review was also performed using PubMed, WOS, and Google Scholar to explore broader UGIB etiologies, organized into five categories:
      1. Mucosal lesions (e.g., infections, radiation-induced damage).
      2. Neoplasms (e.g., GIST, metastases).
      3. Vascular causes (e.g., aneurysms, angiodysplasia).
      4. Bleeding disorders (e.g., coagulopathies, vitamin deficiencies).
      5. External sources (e.g., aortoenteric fistula).
  • Results and Discussion:
    • Gastric Metastases of Breast Cancer (GMB):
      • Rare but impactful, often misdiagnosed as benign conditions (e.g., gastritis).
      • 7 case reports identified; bleeding mechanisms include metastatic lesions, coagulopathies, or varices secondary to liver metastases.
      • Highlights the need for multidisciplinary collaboration and palliative care in advanced cases.
    • Pyloric Gland Adenoma (PGA):
      • 29 case reports reviewed; often misdiagnosed due to nonspecific symptoms (e.g., anemia, obstruction).
      • Endoscopic resection is preferred for small lesions, but large polyps may require surgery.
    • Gastrointestinal Stromal Tumor (GIST):
      • Over 100 publications identified; UGIB is a common presentation.
      • Diagnostic challenges include mimicry of PUD or leiomyomas; immunohistochemistry (e.g., KIT testing) is critical.
  • Diagnostic Pitfalls:
    • Endoscopy misses ~10–20% of UGIB sources; CT/angiography may be needed for elusive lesions (e.g., aortoenteric fistula).
    • Histopathological errors and clinician-pathologist communication gaps can delay diagnosis.
  1. Strengths
  • Comprehensive Approach: Combines systematic case reviews with a narrative synthesis, offering both depth and breadth.
  • Clinical Relevance: Highlights practical challenges (e.g., misinterpretation of imaging/biopsies) and advocates for proactive diagnostics.
  • Case-Based Learning: Uses real cases to illustrate diagnostic workflows and pitfalls.
  1. Limitations

Non-Systematic Methodology: The narrative review lacks PRISMA-style rigor, potentially omitting relevant studies. Even if the PRISMA (Preferred Reporting Items for Systematic Reviews and Meta-Analyses) guidelines are not required for a narrative review because the two serve fundamentally different purposes, when a narrative review includes: “a systematic review of case re-71 ports in the WOS for cases of… overt UGIB of the mentioned causes published between January 2014 75 and May 2025” it is better to use PRISMA statement [1, 2]

  • Selection Bias: Case reports may overrepresent dramatic presentations, skewing perceived rarity/complexity.
  • Generalizability: Focus on proliferative etiologies may underrepresent other rare causes (e.g., vasculitis).
  • In introduction section I suggest the following bibliography (row: 53-55: “Even those 53 common causes can occasionally result from rare underlying conditions, 54 such as Zollinger-Ellison syndrome [3, 4] or pseudocirrhosis [5-7]”
  1. Conclusions and Implications
  • Rare UGIB etiologies are collectively significant and require a high index of suspicion.
  • The review underscores the importance of:
    • Multidisciplinary collaboration.
    • Integrating endoscopic, imaging, and histopathological data.
    • Awareness of systemic conditions (e.g., malignancies) masquerading as localized bleeding.
  • Future Directions: Prospective studies to quantify prevalence and optimize diagnostic algorithms for rare UGIB causes.
  1. Final Assessment

This paper is a valuable resource for clinicians managing UGIB, particularly in complex or atypical cases. Its blend of case reports and narrative review effectively bridges research and practice, though broader systematic reviews could enhance evidence synthesis. The emphasis on diagnostic humility and iterative evaluation is a key takeaway.

  1. REFERENCES:

[1] Page MJ, McKenzie JE, Bossuyt PM, Boutron I, Hoffmann TC, Mulrow CD, et al. The PRISMA 2020 statement: an updated guideline for reporting systematic reviews. BMJ. 2021;372:n71.

[2] Page MJ, Moher D, Bossuyt PM, Boutron I, Hoffmann TC, Mulrow CD, et al. PRISMA 2020 explanation and elaboration: updated guidance and exemplars for reporting systematic reviews. BMJ. 2021;372:n160.

[3] Rossi RE, Elvevi A, Citterio D, Coppa J, Invernizzi P, Mazzaferro V, et al. Gastrinoma and Zollinger Ellison syndrome: A roadmap for the management between new and old therapies. World J Gastroenterol. 2021;27:5890-907.

[4] De Angelis C, Cortegoso Valdivia P, Venezia L, Bruno M, Pellicano R. Diagnosis and management of Zollinger-Ellison syndrome in 2018. Minerva endocrinologica. 2018;43:212-20.

[5] Takata K, Mogi A, Yamauchi R, Shakado S, Hirai F. Pseudocirrhosis Due to Desmoplastic Response to Chemotherapy in Breast Cancer Liver Metastases. Cureus. 2022;14:e25321.

[6] Manenti A, Caramaschi S, Manco G, Bonetti LR. Neoplastic Pseudocirrhosis Pathophysiological Deepening. Dig Dis Sci. 2024;69:4493-4.

[7] Sadlik G, Anderson RC, Lei X, Cen SY, Duddalwar VA, Fong TL. Pseudocirrhosis: A Case Series with Clinical and Radiographic Correlation and Review of the Literature. Dig Dis Sci. 2024;69:1004-14.

Reviewer 2 Report

Comments and Suggestions for Authors

Ion Dina et al.,  Rare Etiologies of Upper Gastrointestinal Bleeding:

The authors had experienced rare entities as causes of UGIB: Gastric metastases of breast cancer (GMB), pyloric gland adenoma, and gastrointestinal stromal tumor (GIST)—we identified 7 and 29 case reports for the first two entities, and over 100 publi- cations addressing GIST.

Search was organized into 5 categories: Lesions of the mucosa, neoplasms, vascular causes, bleeding disorders and external sources of bleeding.

The paper is entertaining and generally well written

Corrections are needed In Table 1, Sistemic Lupus, should be Systemic, Barrett’s esophagus, instead of Barrett esophagus, Barrett’s dysplasia instead of Barrett dysplasia

Mesenchymal instead of mesenchymal

Is Trichobezoar really a cause of bleeding?

Cirrhotic gastropathy instead of cirrhotic coagulopathy, Bleeding is most likely due to portal hypertensive gastropathy and gastric antral vascular ectasy. Is there any reference convincingly showing that cirrhotic coagulopathy causes bleeding, outside or exempt from portal hypertension

Bleeding disorders

It is doubtful to classify Medications induced by thienopyrides and anticogualants as bleeding disorders

I do not think that anticoagulants and thienopyrides are causes of bleeding, but if lesions are present they lead to an increased bleeding

Duodenal papilla instead of papilla, aortic aneurysm, instead of aneurism

Splanchnic instead of splachnic

It would be stronger if the authors could show the references within paranthesis after each etiology in the Table. It would be more convincing, particularly among the rare etiologies

“GISTs are considered rare neoplasms (reported 1.1-1.5 cases per 100.000 person-years in a Scandinavian study)’,…… A reference is missing

Round 2

Reviewer 1 Report

Comments and Suggestions for Authors

Thank You for corrections

Reviewer 2 Report

Comments and Suggestions for Authors

The authors have adequately responded to the comments and criticism